# Prognosis Risk Model Based on Necroptosis-Related Signature for Bladder Cancer

**DOI:** 10.3390/genes13112120

**Published:** 2022-11-15

**Authors:** Zhenghao Chen, Rui Cao, Ren Wang, Yichuan Wang, Donghao Shang, Ye Tian

**Affiliations:** Department of Urology, Beijing Friendship Hospital, Capital Medical University, Beijing 100050, China

**Keywords:** necroptosis, bladder cancer (BLCA), tumor microenvironment (TME), immune checkpoint, immunotherapy

## Abstract

Background: Bladder cancer(BLCA) is the ninth most common cancer. In recent years, necroptosis was found to be related to the occurrence and development of tumors. In this study, we aimed to construct a model based on a necroptosis-related signature to evaluate the potential prognostic application in BLCA. Methods: A total of 67 necroptosis-related genes were used to select the ideal cluster numbers, and it was found that there were four necroptosis-related patterns. Then, we compared the gene expression levels among all of the groups and established a necroptosis-related prognostic model. We made the following enrichment analysis of function and built a novel scoring system, the NEC score, to evaluate the state of necroptosis according to the expression level of necroptosis-related genes. Results: A total of 67 necroptosis-related genes were used to define four distinct necroptosis-related patterns: NEC cluster1–4. Each NEC cluster exhibited different patterns of survival and immune infiltration. Based on univariate Cox regression analyses and least absolute shrinkage and selection operator (Lasso) regression, 14 necroptosis-related genes were established to develop the NEC score. Patients were divided into two groups based on the NEC score. Patients in the high NEC score group had a significantly poorer overall survival than those in the low NEC score group. We further confirmed the correlation of clinical characteristics, as well as the immunotherapy outcome, with the NEC score, and confirmed the potential of the NEC score to be an independent prognostic factor. Furthermore, we compared the expression levels of eight potential biomarker genes between our own BLCA tissues and para-carcinoma tissue. Conclusion: We developed a novel NEC score that has a potential prognostic value in BLCA patients and may help personalized immunotherapy counselling.

## 1. Introduction

Bladder cancer (BLCA) is the ninth most common cancer, and it is a common malignant tumor of the urinary tract [1]. With increasing age, environmental pollution, smoking and other factors, the incidence and mortality of BLCA are continuous rising [2]. Almost four out of five of the first diagnosed BLCA patients are non-muscle-invasive bladder cancer (NMIBC), and half of these patients have a high risk of recurrence. In addition, approximately 20% of these patients’ cancer tends to become muscle-invasive bladder cancer (MIBC) [3]. With the maturity of treatment options such as transurethral resection (TUR), the prognosis of BLCA patients has improved significantly; however, the five-year survival is still merely 60%, 35% and 25% for stage T2, T3 and T4, respectively [4,5]. In recent years, with the progress of molecular pathology, the development of targeted therapies and the revival of immunotherapy, new breakthroughs have been made in the treatment of BLCA, especially for advanced BLCA [6]. Thus, it is essentially necessary to establish a new prognostic model to make targeted therapy, as well as immunotherapy, more precise and flexible.

Necroptosis was first proposed in 2005 by Yuan Junying et al. [7] In their study, the drug necrostatin-1 (Nec-1) was discovered by drug screening, and could inhibit “Caspase-independent necrotic cell death”. The concept of “necroptosis” was formally proposed to distinguish the phenomenon from passive necrotic processes of cells that are not regulated by signals and, in their further research, they confirmed that a feedback loop was formed through FADD and casp8 to limit autophagy, as well as to prevent the rescue pathway from inducing RIPK1-dependent necroptosis, which indicates that RIPK1 could be an essential pathway of necroptosis [8]. In the next few years, research from three different laboratories confirmed the hypothesis. Nader Yatim et al. [9] established models of necroptosis and apoptosis, respectively. They showed that, in dying cells, strong cross-priming requires RIPK1 signaling and NF-κB-induced transcription. The dissociation of NF-κB signaling from inflammatory cell necrosis or apoptosis reduces priming efficiency and tumor immunity. In addition, in Francis Ka-Ming Chan’s review on the crossing of cell death and inflammation, it was also mentioned that the synergy of RIPK3 and its upstream adaptors include RIPK1 and TRIF, as well as DAI. The review also stated that necroptosis promoted inflammation by leaking cellular contents from the damaged plasma membrane [10]. Recent studies have provided a new perspective on the mechanisms of necroptosis, and the relevance in vivo suggested that necroptosis could play important roles in the pathogenesis of various human diseases. In recent years, an increasing number of studies have demonstrated the importance of necroptosis in tumors [11,12,13].

The tumor microenvironment (TME) comprises cellular components, including stromal cells, fibroblasts and endothelial cells, and non-cellular components that exist within and around the tumor. In addition, TME comprises innate and adaptive immune cells. It was reported that macrophages were mainly divided into two subtypes. M1 macrophages can phagocytose tumor cells, whereas M2 macrophages can promote tumor growth and invasion. There is much to explore regarding macrophage-mediated immune modulation and macrophage-mediated drug delivery [14]. Dendritic cells (DCs) are also divided into several subtypes, such as classical DCs, plasmacytoid DCs and monocyte-derived inflammatory DCs. These subsets are also functionally distinct: classical DC1s can cross-present endogenous and exogenous antigens, whereas classical DC2s can only present exogenous antigens. Classical DCs and plasmacytoid DCs are present and active in a steady state, whereas monocyte-derived inflammatory DCs tend to be present only in an inflammatory state. DCs specialize in different functions depending on their status of differentiation [15]. CD8^+^ T cells have the ability to detect specific antigens of tumor cells and play an essential role in the immune process of tumor cells. Various factors are in the way of of CD8^+^ T cells infiltrating, including the activation of oncogenic pathways, tumor hypoperfusion and hypoxia caused by abnormal vasculature, physical barriers constructed by dense stroma and immunosuppressive microenvironments formed by various cells, factors and metabolites [16].

Since we mentioned the role of RIPK1 in inflammation and tumor immunity, the process of necroptosis is usually accompanied by immune activation and an inflammatory response, which is different from apoptosis [17]. Multiple studies indicated that the pro-inflammatory effects of necroptosis play an important role in regulating the TME, and several studies have reported that necroptosis conjointly plays a crucial role in the activity of CD8^+^ T cells [9,18,19]. In addition, Kang YJ et al. demonstrated the RIPK3-PGAM5-Drp1/NFAT signal pathway in the activation of NKT cells in tumors [20], and Lu JV et al. indicated the importance of necroptosis in adaptive and innate immunity via RIPK1 and caspase-8 [21]. Zad-fmk, a pan-caspase inhibitor, had been reported to induce necroptosis, and Werthmöller et al. [22] indicated that the treatment of melanoma with Zad-fmk, combined with other treatments, such as radiation, chemotherapy and hyperthermia, could significantly reduce regulatory T cell (Tregs) infiltration, as well as promote DC and CD8^+^ T-cell infiltration. These phenomena eventually lead to a reduction in tumor growth. Here, we integrated the expression level of necroptosis-related genes for an exploration and comprehensive analysis of necroptosis signatures in BLCA. We found that the necroptosis-related genes of BLCA were associated with vital molecular and clinical features. Based on the necroptosis signature, we defined four necroptosis patterns associated with different overall survival (OS) and TME, suggesting that necroptosis played an essential role in forming the TME of BLCA patients. Therefore, we constructed a new score system, the NEC score, that quantified the necroptosis related gene status of patients. The NEC score can predict the prognosis, immune infiltration and immunotherapy response of BLCA. Finally, we evaluated eight key genes of the NEC score in our own patients’ sample by qPCR. Our findings revealed a potential relationship between necroptosis and the prognosis and TME, as well as the effect of immunotherapy in patients with BLCA.

## 2. Materials and Methods

### 2.1. Data Collection and Processing

We searched available gene expression datasets, including four datasets (GSE32548, GSE13507, GSE48075, and GSE32894) and TCGA-BLCA cohort, for BLCA, and 995 samples were finally collected and merged to be the meta-BLCA cohort in this study. All clinical information from the datasets was acquired from the GEO and TCGA database. We downloaded the level of RNA-sequencing data for FPKM of TCGA-BLCA dataset from the TCGA GDC (https://portal.gdc.cancer.gov/) accessed on 10 September 2021, and FPKM values were transformed into transcripts per kilobase million (TPM) [23]. Clinical data of the TCGA-BLCA cohort were obtained from UCSC Xena and the supplemental information was obtained from the research of Robertson et al. [24]. Details of the clinicopathological features of each dataset were also included in our previous study [25]. The mutation data of TCGA-BLCA cohort were acquired from the GDC [26]. The total number of mutations counted in the whole exon territory of each sample was calculated as the TMB according to a previous study [27]. Moreover, our study also included the IMvigor210 (mUC) dataset of mUC patients receiving PD-L1 inhibitor atezolizumab to validate results. The raw transcriptomic and clinical data were obtained from the IMvigor210 (mUC) dataset [28]. The raw count was also transformed to TPM to represent gene expression in the IMvigor210 (mUC) dataset. Data were analyzed with the R (version 4.0.5) and R Bioconductor packages.

### 2.2. Consensus Clustering for Necroptosis-Related Patterns

The generally accepted genes of necroptosis are listed in Appendix A. To identify distinct necroptosis-related patterns (NEC cluster), the consensus clustering method (K-means) was applied [29]. The number of clusters in meta-BLCA cohort was determined by consensus clustering algorithm. This process was repeated 1000 times to ensure the stability of classification using the ConsensusClusterPlus package in R. 

### 2.3. Identification of Differentially Expressed Gene between Different Necroptosis-Related Patterns

The differentially expressed gene of each necroptosis-related patterns was determined by limma package in R [30]. The significance criteria were set as |log2 fFC| > 1.0 and FDR < 0.05.

### 2.4. Evaluation of Infiltrating Immune Cells in the TME

In our study, we used the ssGSEA to evaluate the infiltrated level of immune cells in the samples. We acquired and merged the marker gene sets for TME infiltration immune cell types from the studies of Bindea et al. [31]. and Charoentong et al. [32]. The normalized enrichment score was used to represent the relative amount of each immune cell infiltrating in BLCA. 

### 2.5. Functional and Pathway Enrichment Analyses

The differences in biological processes among each NEC cluster were explored by gene enrichment analyses through clusterProfiler package in R [33]. We curated a series of gene sets to represent specific biological processes, which was constructed by Mariathasan et al. [28,34]. Correlations between distinct NEC clusters and the biological pathways were further performed.

### 2.6. Establishment of NEC-Cluster-Related Gene Signature

We built up a NEC-cluster-related gene scoring system, NEC clusterNEC score, to evaluate the necroptosis-related genes expression status of individuals. The procedures establishing the NEC score were as mentioned before [35]. To reduce noise or redundant genes, we used the Cox regression model with LASSO, and the NEC score of each sample was calculated by the relative expression of NEC-cluster-related genes and its Cox coefficient.

### 2.7. Tissue Specimens 

A total of 55 BLCA specimens and paired para-carcinoma tissue samples were obtained at the Beijing Friendship Hospital, Capital Medical University (Beijing, China) between January 2021 and March 2022 as previously described. The clinical characteristic of the patients was listed in Appendix A. The clinical BLCA specimens were collected with permission from our Institutional Research Ethics Committee.

### 2.8. RNA Extraction, Reverse Transcription and Quantitative Real-Time PCR (qRT-PCR)

The RNeasy plus mini kits (Qiagen, Stockach, Germany) were used to extract total RNA from clinical samples according to manufacturer’s instructions. Subsequently, the quality of the extracted RNA was detected with NanoDrop instrument (Implen, Munich, Germany). Then, the RNA was used as a template for cDNA synthesis using the ReverTra Ace qPCR RT Kit (Toyobo, Tokyo, Japan). Finally, forward and reverse primers of our target genes and iQTM SYBR ® Green Supermix (Bio-Rad, Beijing, China) were mixed, and the qRT-PCR was performed. The primer sequences were listed in Appendix A. The relative expression level of the targeted genes was normalized to GAPDH.

### 2.9. Statistical Analyses

Student’s *t* tests were used to test the statistical significance between two groups, and, for those variables divided in more than three groups, one-way ANOVA or Kruskal–Wallis tests were used depending on the types of data. The correlations between the NEC score and clinical parameters were analyzed by Thec2tes. Survival rates were calculated and visualized by KM survival curves, and the significant differences were tested with the log-rank test. Pearson correlation analyses were used to calculate the correlation coefficients between immune cells, as well as distinct gene sets, and the NEC score. HRs of variables and whether those variables were independent prognostic factors were determined by univariate and multivariate Cox proportional hazard models. A nomogram and calibration curves were formed, and, according to the suggestion of Iasonos et al. [36], DCA was used to determine whether our nomogram was suitable for clinical use. We used waterfall plot to visualize the mutation status of patients with distinct NEC cluster in meta-BLCA cohort using packages maftools [37] and complexheatmap [38] in R. Significance of statistical analysis was set as *p* < 0.05.

## 3. Results

### 3.1. Characterization of Necroptosis-Related Genes Expression in BLCA

The scheme of the necroptosis-related genes expression of the NEC cluster is shown in Figure 1A. We used the Consensus Cluster Plus package to select the cluster numbers and found four distinct necroptosis-related clusters (NEC clusters) in the meta-BLCA cohort, including 71 cases in cluster 1, 267 cases in cluster 2, 351 cases in cluster 3 and 305 cases in cluster 4, which were named as NEC cluster1–NEC cluster4, respectively (Figure 1B). Hierarchical clustering revealed a differential expression of the necroptosis-related genes in NEC cluster1–NEC cluster4 in the meta-BLCA cohort (Figure 1C). Furthermore, there were significant differences in the prognosis of these four clusters of patients according to Kaplan–Meier survival curves (log-rank test, *p* < 0.00001; Figure 1D), and NEC cluster1 and NEC cluster3 exhibited the better survival advantage comparing with other patterns.

### 3.2. The TME Immune Cell Infiltration in Distinct NEC Cluster

To investigate the relationship between TME and NEC clusters, we used the ssGSEA algorithm to calculate the relative numbers of immune cells infiltrating the TME. A cluster heatmap analysis revealed that there were different different TME immune cell infiltrations among distinct NEC clusters (Figure 2A). A significant increase in mast cells, DCs, B cells and cytotoxic cells was found in NEC cluster1; however, the most notable feature of NEC cluster1 was that the CD8 T cells were not as highly expressed as the other immune cells (Figure 2A,C). NEC cluster2 had an extensive infiltration of immune cells, including DCs, B cells, macrophages, CD8 T cells and cytotoxic cells, but there was also a substantial infiltration of NEC cluster2 regulatory T cells. After a significant infiltration of immune cells, Treg cells respond to this phenomenon by increasing their numbers. The immune cell infiltration of NEC cluster3 was similar to that of NEC cluster1, except for the expression of CD8 T cells and cytotoxic cells. NEC cluster3 was characterized by obvious highly expressed CD8 T cells, as well as other important immune cells. NEC cluster4 showed a low immune cell infiltration, lacking almost all of the immune cells, which explained the terrible prognosis of NEC cluster4.

### 3.3. Characteristics of the Biological Process in NEC Clusters

By running ssGSEA on the Kyoto Encyclopedia of Genes and Genomes (KEGG) pathway, we investigated biological processes in four different ways. As shown in Figure 2. NEC cluster1 and NEC cluster3 were basically activated in different pathways. NEC cluster1 was activated in angiogenesis as well as the KRAS DN pathway, and, in addition, almost all of the dead patients were in the condition of epithelial mesenchymal transition (EMT) activation. NEC cluster2 was markedly activated in the immune pathway, including the activation of the interferon-gamma response and interferon-α response. Previous studies have shown that the pan-cancer immune cell infiltration pattern divides cancer patients into three main subtypes: immune desert type, immune rejection type and immune inflammatory type. The immune rejection phenotype is characterized by massive immune cell infiltration into the stroma surrounding the central tumor niche [39]. We had already found above that the TME immune cell infiltration of NEC cluster2 was remarkably increased; thus, the highly activated angiogenesis, as well as the EMT pathway, also proved our hypothesis that NEC cluster2 was a typical immune-excluded phenotype, and that NEC cluster3 markedly presented pathway enrichment associated with pathways in cancers, including the transforming growth factor (TGF)-β signaling pathway, WNT signaling pathway and P53 signaling pathways. In addition, most pathways in NEC cluster4 were not activated, except some cell-cycle-related pathway and DNA-damage-repair-related pathways, such as the hallmark g2m checkpoint, hallmark unfolded protein response (UPR) and hallmark DNA repair. NEC cluster3 significantly enriched cancer-related signaling pathways, including transforming growth factor (TGF)-β signaling, WNT signaling and P53 signaling. Furthermore, we noticed a massive infiltration of CD8^+^ T cells in NEC cluster3 patients, which is considered to be one of the most important parts of cellular immunity [40]. In addition, we also discovered that the expression levels of major histocompatibility complex (MHC) molecules, which were responsible for T cell activation, were comprehensively elevated in NEC cluster3, whereas those in NEC cluster1 were inhibited, indicating that NEC cluster3 might be better candidates for immunotherapy (Figure 2D).

### 3.4. Establish of the NEC Score

Although necroptosis-related genes played important roles in different TME statuses, predicting individual necroptosis-related patterns were not convenient considering that all of the above analyses were performed in populations. Given the heterogeneity and complexity of individual necroptosis-related gene expression, we aimed to create a scoring system to quantify individual necroptosis-related genes in BLCA, termed the NEC score.

To describe necroptosis-related expression patterns through transcriptome data, based on univariate Cox regression analysis, 159 genes that were differentially expressed between the subtypes associated with prognosis were finally defined as prognostic candidate genes associated with necroptosis (Appendix A). A regression analysis of these genes using the least absolute shrinkage and selection operator (LASSO) has established a signature (NEC score) to better characterize necroptosis-related genes. Finally, 14 genes were recruited to establish the NEC score, and the coefficients of these genes are listed in Appendix A.

Then, to determine the predictive value of the NEC score for patient prognosis, patients were divided into low and high NEC score groups according to the median. KM survival curves showed a higher survival in patients with lower NEC scores (log-rank test, *p* < 0.001; Figure 3A). The Kruskal–Wallis test indicated a significant difference in the NEC score among NEC clusters; it is worth noting that the harmful NEC cluster2 and NEC cluster4 showed a higher NEC score whereas the beneficial clusters, NEC cluster1 and NEC cluster3, showed a lower NEC score (*p* < 0.001; Figure 3B). 

A molecular characteristic has been shown by TCGA, as well as other groups, and these groups divided BLCA into different molecular subtypes. Thus, we evaluated the correlation of the NEC score and these molecular subtypes (Appendix A), taking Lund as an example. It is worth noting that we found that patients with a low NEC score prefer to be in subtypes of urothelial-like A (UroA), as well as genomically unstable (GU), which were both generally considered to be associated with lower malignancy and better survival (Figure 3C). The low NEC score group had over six times more UroA patients than the high NEC score group (46.3% vs. 7.1%), GU patients in the low NEC score group were more than the high NEC score group (25.4 %VS 18.4%), and the number of basal SCC-like patients in the high NEC score group was larger than the low NEC score group (46.9% vs. 8.2%). These results indicated that ICIs in the patients in the high NEC score group might be less efficient (Figure 3D). Meanwhile, basal/SCC-like molecular subtypes, as well as urothelial-like B (UroB), which are usually associated with a high malignancy, as well as a terrible prognosis, were gathered in the high NEC score group (Figure 3C). We further performed function annotation to illustrate the characteristics of the NEC clusters. The results showed that, with the increase in the NEC score, the CD8 T cell as well as CD56bright natural killer cells showed an obvious downward trend, whereas the NEC score was significantly positively correlated with Treg cells. (Figure 3E). An alluvial diagram was then used to visualize changes in the distribution of individual molecular subtypes (Figure 3F).

### 3.5. NEC Score Can Be Used as an Independent Prognostic Factor in BLCA

Since the NEC score is associated with malignancy and survival, we aimed to determine whether the NEC score was an independent prognostic factor in BLCA patients using univariate and multivariate Cox regression analysis. The NEC score was analyzed as a covariate with other clinical characteristics (e.g., age, T stage, N stage). The results indicated that the age, T stage, N stage and NEC score were independent prognosis predictors (Figure 4A,B). We constructed a nomogram as a clinically relevant quantitative method by combining independent prognostic factors. Clinicians can use this nomogram to predict the prognosis of BLCA patients (Figure 4C). By adding up the scores of each parameter, each patient will be assigned a total score. Higher overall scores correspond to poorer patient outcomes. Decision curve analysis (DCA) also indicated that the nomogram had great potential application value (Figure 4D).

Previous studies have shown that ICI-based immunotherapy is undoubtedly a major breakthrough in cancer treatment. We followed this up to explore whether the NEC score is applicable to the IMvigor210 (mUC) cohort, and to explore whether the use of the IMvigor210 (mUC) cohort was predictive of the benefits of immunotherapy for BLCA. KM survival curves revealed that patients with a low NEC score had a significantly better survival when compared with the high NEC score group (log-rank test, *p* = 0.00387, Figure 5A). Though there were no differences in CD8^+^ T cells between these two groups, an obviously higher expression of regulatory T cells could significantly affect the prognosis of patients in both groups (Figure 5B). In addition, the correlation matrix also showed that the NEC score was positively correlated with regulatory T cells, whereas it was negatively correlated with T cells, DCs and T helper cells, which were all considered as negative factors of immunosuppression (Figure 5C). Meanwhile, we also verified the predictive value of our NEC score in TCGA-cohort, GSE13507, as well as GSE32548, respectively (Appendix A).

In addition, in our results, both GU subtypes and TCGA II subtypes had relatively low NEC scores (Figure 5D,E), which indicated that the patients in the low NEC score group were more likely to respond to immunotherapy. This is in line with our previous findings in the meta-BLCA cohort in Figure 3.

Furthermore, we also found that the TMB and NEC score were negatively correlated in the IMvigor210 (mUC) cohort (*p* = 0.02, Figure 5F). The Kaplan–Meier survival curves showed that the survival rate of the low NEC score and high TMB group was higher than that in the rest groups; in addition, regardless of the TMB level, patients with a lower NEC score seemed to have a better prognosis compared with those who had a higher NEC score (Figure 5G). Moreover we found that the expression of MHC in high/low NEC score groups was relatively different (Figure 5H). Additionally, patients with a lower NEC score tend to benefit more from immunotherapy (Wilcoxon test, *p* = 0.017, Figure 5I), confirming earlier conclusions that the NEC score is a predictor of immunotherapy. 

### 3.6. Expression Level of NEC Score-Related Candidate Genes in BLCA Patients’ Sample

We further validated the expression of 8 out of 14 NEC-score-related candidate genes in BLCA patients’ samples by qPCR. These eight genes were screened by combining the expression in tumor and tumor-adjacent tissue, as well as the correlation between gene expression and staging/survival in the BLCA cohort of the GEPIA2 website (GEPIA 2 (cancer-pku.cn)). Unsurprisingly, the expression of these eight genes was basically in line with the conclusions of GEPIA and our results. 

The expression of CSE1L, LOX, RHBDD1, RRBP1 and SOGA1 was significantly higher in tumors than in tumor-adjacent tissue, and the survival analysis of each gene reached the same conclusion (Figure 6). Meanwhile, the expression of PALLD and GEMIN2 in the tumor was obviously lower than that in tumor-adjacent tissue. (Figure 6). However, AHNAK showed no significantly difference between the tumor and tumor-adjacent tissues that we collected; a further large sample examination needs to be carried out. In conclusion, our findings suggest that necroptosis-related subtypes are significantly associated with the immunophenotype and that the NEC score helps to predict the patient response to the ICI immunotherapy NEC score.

## 4. Discussion

BLCA is a common malignant tumor of the urinary system. BLCA is the ninth leading cause of cancer-related death worldwide due to its high recurrence rate and propensity for metastasis [41]. In recent years, increasing transcriptomic data from public databases such as TCGA and GEO are easily accessible. Therefore, analyzing and evaluating genomic and transcriptomic sequencing data in various human cancers can provide us with a comprehensive view of carcinogenesis in terms of genetic, epigenetic and protein-level aberrations [42]. In recent years, cancer research has gradually shifted from focusing merely on tumor cell metastasis to surrounding core cancer cells, termed TME [15,43,44]. Key components of the TME, including stromal cells, immune cells and chemokine-secreting cells, synergistically create a chronic, inflammatory, immunosuppressive and tumor-promoting environment that protects cancer cells from stringent immune surveillance [43]. During cancer progression, tumor and microenvironment cells are constantly in cellular stress and damage, as well as an infected state, which is the main cause of necroptosis, a type of programmed cell death. The occurrence of tumor tissue necroptosis is induced by the abnormal TME in tumors, which, in turn, affects immune function. 

Here, we explored the expression profiles from TCGA, GEO and the IMvigor210 (mUC) cohort, and a comprehensive evaluation of necroptosis signatures in BLCA. According to the expression of necroptosis genes, we divided BLCA patients into four subtypes through the consensus cluster and found significant differences in gene expression characteristics, survival and immune infiltration among the groups. Through the differential genes among the subtypes, we used lasso regression to establish the necroptosis signature and calculated the NEC score. Furthermore, we found a correlation between the NEC score and the survival prognosis, molecular subtypes and immune cell infiltration in the TCGA cohort, as well as the IMvigor210 (mUC) cohort. In addition, the predictive ability of the NEC score for immunotherapy has also been proven. We confirmed our results in the patients’ samples that we collected and found the potential biomarker genes CSE1L, LOX, RHBDD1, RRBP1, SOGA1, PALLD and GEMIN2. In a previous study, by examining tumor biopsies collected from patients before treatment, three basic immune signatures, termed as immune-inflamed, excluded and desert, were distinguished in relation to anti-PD-L1/PD-1 therapy [39]. The first subtype is the immune-inflamed phenotype, which is characterized by the substantial presence of both CD4^+^ and CD8^+^ T cells in the tumor, often with myeloid cells and monocytes; the immune-related cells are in the vicinity of the tumor cells. The second subtype, immune-excluded, is also characterized by the presence of a large number of immune cells. However, immune cells did not invade the parenchyma of these tumors, but remained in the stroma surrounding the tumor cell nests. The stroma can be trapped in the tumor envelope or can penetrate the tumor itself, suggesting that immune cells are indeed present within the tumor. A third immune-desert subtype is characterized by the absence of T cells. Although myeloid cells may be present, it is generally characterized by the presence of a non-inflammatory TME with few or no CD8-carrying T cells [39].

In our study, NEC cluster4 was regarded as an immune-desert phenotype with a significantly low immune cell expression, especially CD4^+^ and CD8^+^ T cells. NEC cluster2 was characterized by abnormally increased whole immune cells, and stromal pathways were also markedly activated (Appendix A). This indicated that NEC cluster2 was considered to be in the typical immune-excluded subtype. 

Furthermore, NEC cluster3 was characterized by highly expressed immune cells, especially CD8^+^ T cells; considering the low expression of the stromal-related pathway, we believed NEC cluster3 to belong to the immune-inflamed phenotype. As for NEC cluster1, we believed that it belonged to a special subtype between immune-inflamed and immune-excluded. On the one hand, there was a large amount of immune cell infiltration, except CD8^+^ T cells, and, on the other hand, the expression of the stromal pathway level, especially the angiogenesis, was greatly increased. Also concerning the CD8^+^ T cells in NEC cluster1, Daniel S. Chen explained in their review that CD8^+^ lymphocytes in tumors may also exhibit dysfunctional states, such as excessive exhaustion. Tumor cells in inflammatory tumors can also express inhibitory factors that downregulate the expression level of MHC class I molecules to desensitize anticancer immunity [39], which could be explained with CD8^+^ T cell exhaustion in cancer [45,46].

Furthermore, DEGs have been reported to be enriched in different groups in stromal and immune-related biological processes, and have been identified as genes associated with NEC clusters. Based on this, we established a new NEC-cluster-associated genetic marker system to assess necroptosis in BLCA individuals. In the meta-BLCA and IMvigor210 (mUC) cohorts, we found that patients with higher NEC scores had a poorer prognosis. In addition, the NEC score of each subtype also confirmed its positive correlation with survival. 

According to several ICI clinical trials, the immune cell infiltration status in the TME is now considered to be very valuable and important information for predicting the prognosis and response to various cancer immunotherapies [47,48,49,50]. Therefore, we calculated a comprehensive analysis of TME immune cell infiltration by estimating the infiltration levels of various types of immune cells in BLCA. Unsurprisingly, most of the immune cells that were beneficial to immune activity were activated in the low NEC score group, whereas those immune cells detrimental to immune activity were positively correlated with the NEC score. Meanwhile, in the IMvigor210 (mUC) cohort, the Treg level of the high NEC score group was significantly higher, whereas the expression of CD8^+^ T cells in the high NEC score group was significantly lower. This further indicated why the high NEC score group has a worse prognosis than the low NEC score group. As our NEC cluster was able to differentiate between different TME immune cell infiltrations and immunophenotypes, we further analyzed whether the NEC cluster that we established was somehow related to TMB, as the mutational landscape is considered as a potential biomarker used to predict patients’ clinical response to immunotherapy [51]. Our results showed that patients who had a higher TMB tended to have a lower NEC score. Furthermore, according to the Kaplan–Meier survival curves, considering both the NEC score and TMB, it could clearly be found that patients in the low NEC score and high TMB group had a significantly better prognosis than the other groups. In contrast, patients in the high NEC score and low TMB group had a significantly worse prognosis than the other groups. This result was consistent with clinical experience.

Regarding the key genes, we screened them and proved them to be differently expressed by qPCR for the NEC score, including CSE1L, LOX, RHBDD1, RRBP1 SOGA1, PALLD and GEMIN2. There were two groups that had the higher expression of CSE1L in BLCA patients, which was proven by immunohistochemical analysis [52,53]. LOX is widely known to catalyze the covalent crosslinking of collagens and elastin in the extracellular matrix (ECM), an essential process for the structural integrity of all tissues. LOX enzymes can also remodel the TME and have been implicated in all stages of tumor initiation and the progression of many cancer types. Changes in the ECM can influence several cancer cell phenotypes [54]. Matsuyama M et al. proved the inhibition of cell proliferation in urological cancer cell lines, including renal cell carcinoma (RCC), BLCAand prostate cancer (PC), by using lipoxygenase inhibitors [55]. The effect of RHBDD1 on BLCA is rarely studied, but its cancer-promoting role in colorectal cancer was confirmed in the study by Zhang M et al. [56]. As for RRBP1, it was proven to be highly expressed in BLCA, and the correlation with migration and invasion was verified by various groups [57,58,59]. In addition, the effect of PALLD on BLCA is rarely studied, but Pogue-Geile KL et al. proved that the palladin mutation could cause the tumor’s strong invasive and migratory abilities in pancreatic cancer [60]. Last but not least, the mechanistic studies of GEMIN2 and SOGA1 in tumors are very few at present, and they are also the main subjects of our follow-up research.

Furthermore, patients with higher NEC scores tended to have higher malignant clinicopathological features and were associated with more aggressive molecular subtypes, whereas the opposite phenomenon was found in the low NEC score group. Furthermore, patients with genomic instability and the TCGA II subtype, who tended to be more susceptible to immunotherapy [61,62], were correlated with a low NEC score in the IMvigor210 (mUC) cohort. On the other hand, patients with TCGA subtype III/IV, characterized by stromal activation and immunotherapy, were significantly associated with a higher NEC score (Figure 5D–G).

## 5. Conclusions

Our NEC score is a reliable, comprehensive assessment of necroptosis-related subtypes in BLCA patients, correlated with clinical, cellular and molecular features, including the clinical stage, TME immune cell infiltration, molecular subtype, genetic variation and tumor mutation status. In addition, the NEC score can be used as an independent prognostic factor for the prognosis of BLCA patients and a predictor of the clinical response to immunotherapy. We also confirmed the expression levels of the key NEC score genes in our own samples. Furthermore, this study provides a solid theoretical basis for the study of the role of necroptosis-related signaling pathways in BLCA and their subsequent clinical application.

## Figures and Tables

**Figure 1 genes-13-02120-f001:**
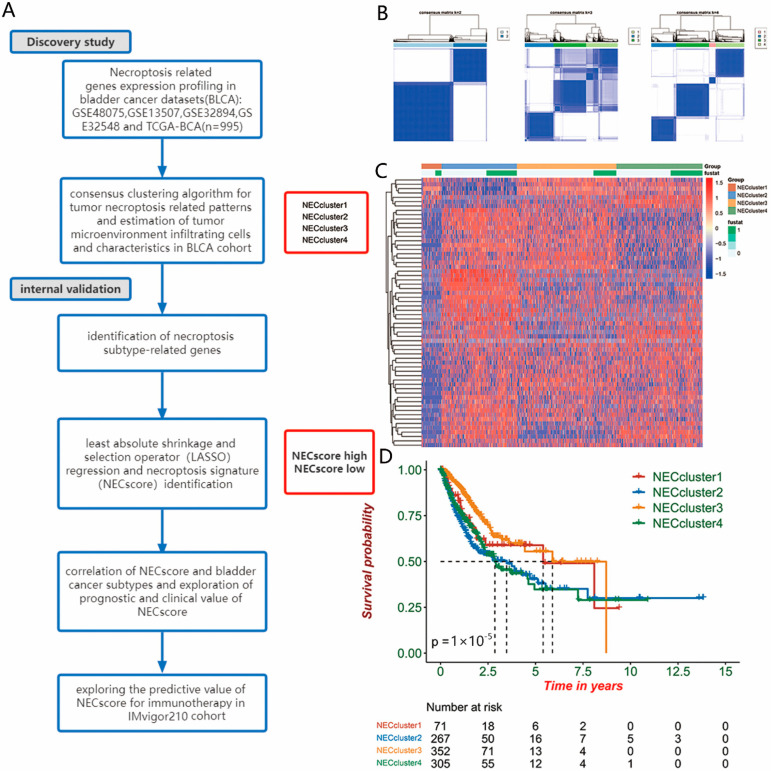
Consensus clustering of necroptosis-related genes in bladder cancer (BLCA). (**A**) Overview of study design. (**B**) Consensus matrices of patients in the BLCA cohort for k = 2–4 using 1000 iterations of unsupervised consensus clustering method (K-means) to ensure the clustering stability. (**C**) Hierarchical clustering of necroptosis-related genes in meta-BLCA cohorts. Hierarchical clustering was performed with Euclidean distance and Ward linkage. The NEC clusters and BLCA cohorts are shown as patient annotations. Rows represent necroptosis-related genes, and columns represent BLCA samples. Red represents high expression and blue represents low expression of each gene. (**D**) Survival analyses for the distinct necroptosis-related genes expression in the meta-BLCA cohort using Kaplan–Meier curves. The NEC cluster1 and NEC cluster3 showed significantly better OS than the other two clusters (log-rank test, *p* < 0.0001).

**Figure 2 genes-13-02120-f002:**
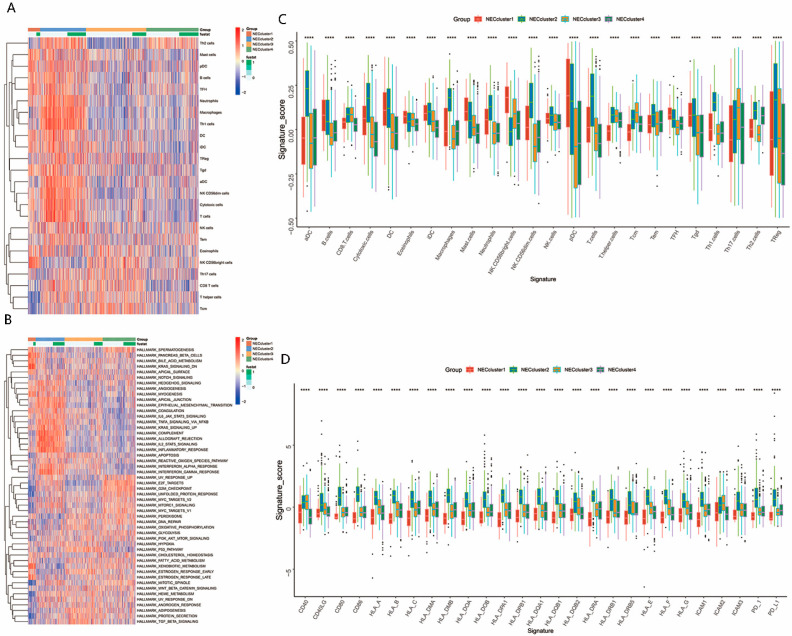
TME immune cell infiltration and biological process characteristics in distinct hypoxia response patterns. (**A**) Hierarchical clustering of TME infiltration immune cells for 403 patients in the meta-BLCA cohort. Rows represent relative amount of each immune cell, and columns represent BLCA samples. Red represents relative upregulation and blue represents relative downregulation of each immune cell. The NEC clusters are shown as patient annotations. (**B**) ssGSEA showed the relative activity of biological pathways in distinct necroptosis-related gene expression patterns in the meta-BLCA cohort. The heatmap was used to visualize indicated biological processes, including immune activation, antigen processing, mismatch repair, pathway in cancers, etc. Red represents that the pathways were relatively activated and blue represents that the pathways were relatively suppressed in each sample. The NEC clusters were used as sample annotations. (**C**) Difference in the enrichment of indicated signatures to represent specific biological processes, including stromal-activation-relevant signatures, mismatch-repair-relevant signatures and immune-activation-relevant signatures among four distinct necroptosis-related genes expression patterns in the meta-BLCA cohort. (**D**) Differences in the expression levels of MHC molecules, costimulatory molecules and adhesion molecules among four distinct necroptosis-related gene expression patterns in the meta-BLCA cohort. In (**C**) and (**D**), the upper and lower ends of the boxes represent the interquartile range of values. The lines in the boxes represent the median value and the black dots show outliers. The statistical difference among four distinct necroptosis-related genes expression patterns was tested by a one-way ANOVA test. **** *p* < 0.0001.

**Figure 3 genes-13-02120-f003:**
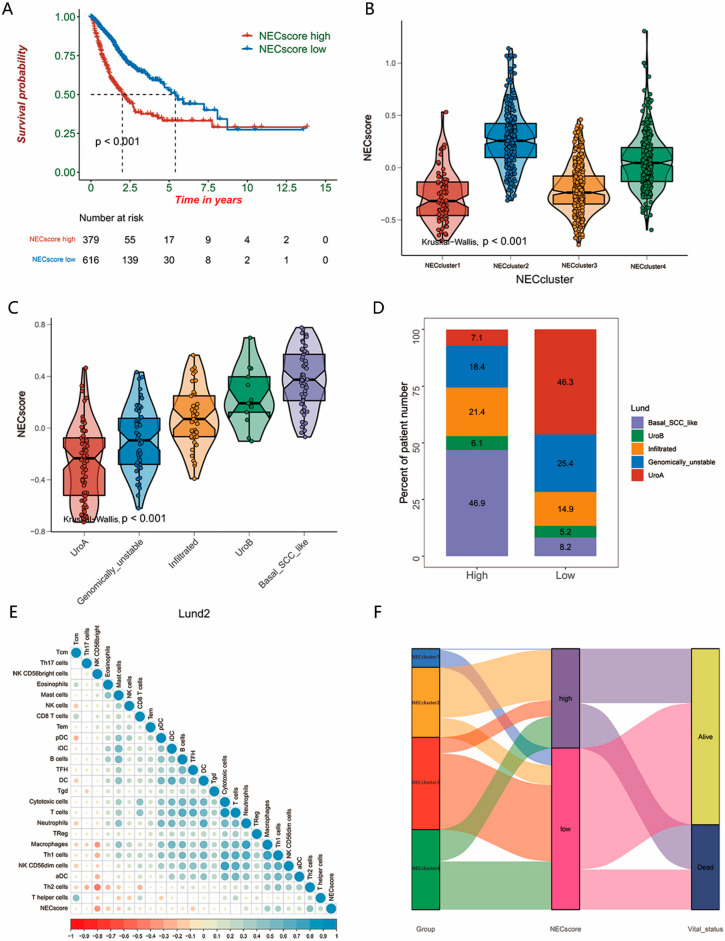
Transcriptome traits and biological characteristics of necroptosis-related gene signature (NEC score) in meta-BLCA cohort (**A**) Kaplan–Meier survival curves showed the difference in prognosis advantage between high and low NEC score groups in meta-BLCA cohort (log-rank test, *p* < 0.001). (**B**) Distributions of NEC score in four distinct necroptosis-related gene expression patterns. Kruskal–Wallis test was used to compare the statistical difference between each pattern (*p* < 0.001). (**C**) Differences in NEC score between different NEC clusters in the Lund classification system. The Kruskal–Wallis test was used to compare the statistical difference between different molecular subtypes (*p* < 0.001). (**D**) The proportion of Lund molecular subtypes in high and low NEC score groups in the meta-BLCA cohort. (**E**) Correlation between NEC score and gene signatures linked to immune and stromal activation in meta-BLCA cohort. Negative correlation is marked with blue and positive correlation is marked with red. (**F**) Alluvial diagram showing the changes in NEC score, NEC clusters and vital status in meta-BLCA cohort.

**Figure 4 genes-13-02120-f004:**
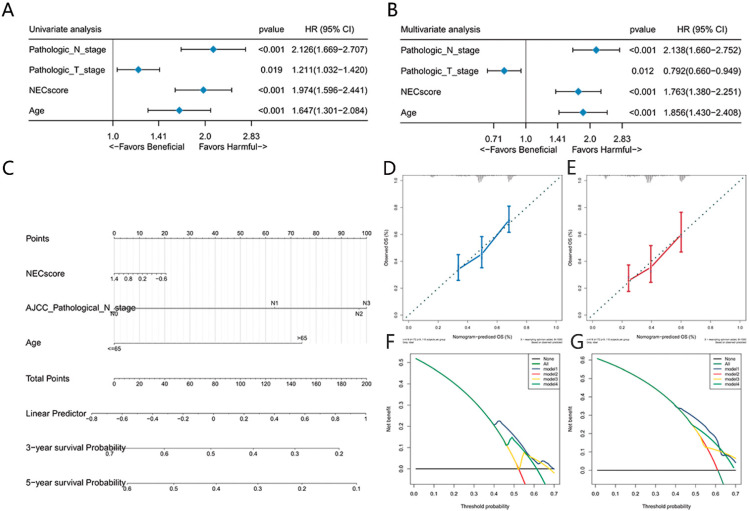
NEC score is an independent prognosis factor in the nomogram. (**A**,**B**) Forest plot summary of the univariate (**A**) and multivariate (**B**) Cox analyses of the NEC score and clinicopathological characteristics. The results indicate four independent prognosis factors: age, pathologic T stage, pathologic N stage and NEC score. The blue diamond squares on the transverse lines represent the HR, and the black transverse lines represent the 95% confidence interval (CI). The p value and 95% CI for each clinical feature are displayed in detail. (**C**) Nomograms for predicting the probability of patient mortality at 3 or 5 year OS based on five independent prognosis factors. (**D**,**E**) Calibration curves of the nomogram for predicting the probability of OS at 3 and 5 years. (**F**,**G**) Decision curve analyses (DCAs) of the nomograms based on five independent prognosis factors for 3-year and 5-year risk.

**Figure 5 genes-13-02120-f005:**
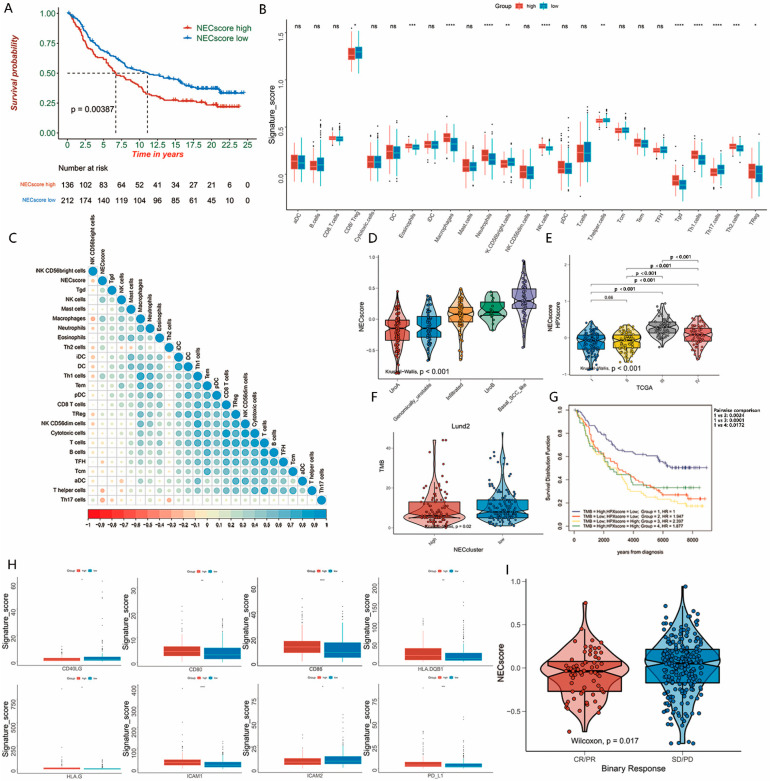
Characteristics of NEC score in IMvigor210 (mUC) cohort. (**A**) Kaplan–Meier survival curves show the difference in prognosis advantage between high and low NEC score groups in the IMvigor210 (mUC) cohort (log-rank test, *p* = 0.00387). (**B**) Difference in the enrichment of the indicated gene signature between high and low NEC score groups in the IMvigor210 (mUC) cohort. The upper and lower ends of the boxes represent the interquartile range of values. The lines in the boxes represent median value, and the black dots show outliers. The statistical difference between high and low NEC score groups was tested by the Student’s *t* test. ns non-significant, * *p* < 0.05, ** *p* < 0.01, *** *p* < 0.001, **** *p* < 0.0001. (**C**) Correlation between NEC score and indicated gene signature in IMvigor210 (mUC) cohort. Blue indicates negative correlation and red indicates positive correlation. (**D**) Differences in NEC score among different molecular subtypes in the Lund classification system in IMvigor210 (mUC) cohort. The Kruskal–Wallis test was used to compare the statistical difference between different molecular subtypes (*p* < 0.00001). (**E**) Differences in NEC score among different molecular subtypes in the TCGA cluster classification system in IMvigor210 (mUC) cohort. The Kruskal–Wallis test was used to compare the statistical difference between different molecular subtypes (*p* < 0.00001). (**F**) Difference in TMB between high and low NEC score groups in the IMvigor210 (mUC) cohort. The TMB was log2 transformed. The statistical difference was measured with the Wilcoxon test (*p* = 0.02). (**G**) Kaplan–Meier survival curves show the difference in prognosis advantage among four groups stratified by NEC score and TMB in the IMvigor210 (mUC) cohort. (**H**) Difference in the expression of immune checkpoints and immune activation-related genes between high and low NEC score groups in the IMvigor210 (mUC) cohort. (I) Differences in NEC score between different ICI immunotherapy clinical response groups. The statistical difference was measured with the Kruskal–Wallis test (*p* = 0.017).

**Figure 6 genes-13-02120-f006:**
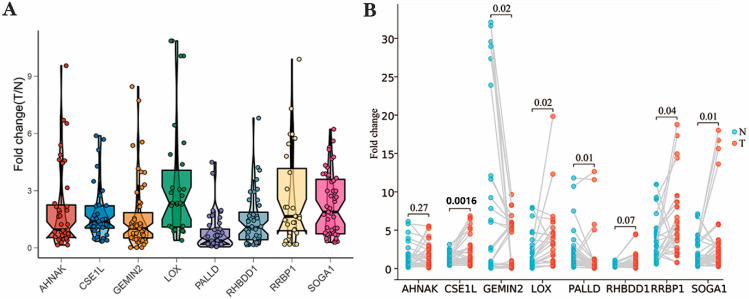
Expression levels of the potential biomarker genes. (**A**) Fold change in expression of potential biomarker genes AHNAK, CSE1L, GEMIN2, LOX, PALLD, RHBDD1, RRBP1 and SOGA1. The statistical difference was measured with Kruskal–Wallis test (*p* < 0.00001). (**B**) Fold change in expression of potential biomarker genes of each pair of tumor and para-carcinoma tissue. The statistical difference was measured with the paired Students’ *t* test.

## Data Availability

The datasets supporting the conclusions of this article are included within the article and Appendix A.

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
