# Peer review of "Prognosis Risk Model Based on Necroptosis-Related Signature for Bladder Cancer"

_genes, 2022, doi:10.3390/genes13112120_

Round 1

Reviewer 1 Report

Prognosis Risk Model Based on Necroptosis-Related signature for Bladder Cancer” by Zhenghao Chen et al describes the potential and the essential of necroptosis-related genes or NECscore to predict the response to immunotherapy as well as be an independent prognosis predictor in balder cancer. The manuscript is written very well. I have several questions that need to be clarified.

1.     About the molecular subtype mentioned in the manuscript, why did you use the Lund classification system? In the TCGA-BLCA (2017 database) the research team pointed out five molecular subtypes: luminal-papillary, luminal-infiltrated, luminal, basal-squamous, and neuronal.

2.     Did you check the correlation between NECscore and PD-L1 expression? 

Author Response

We really appreciate your time to review our manuscript and put forward valuable and constructive comments. Please see the answer to the comments the attachment.

Reviewer 2 Report

Chen et al analyzed microarray datasets from GSE13507, GSE32548, GSE32894, GSE48075 and TCGA-BLCA for BLCA. From tissue samples, a total of 67  necroptosis-related genes were investigated in this study. The results indicated four distinct patterns, which are closely linked with necroptosis (NEC) clusters.

Based on this pattern, these clusters showed different patterns of immune cells infiltration and survival benefits.  

Furthermore, using univariate Cox regression analyses and least absolute shrinkage and selection operator (Lasso) regression, 14 necroptosis-related genes were established to develop NEC score.

Notably, differential gene expressions of CSE1L, LOX, RHBDD1, RRBP1 SOGA1, PALLD and GEMIN2 were reported in this study. 

Survival analyses indicated poor survival was directly correlated with high NEC score. Clinical and immunotherapy outcomes also confirmed NEC score can be independent prognostic for BLCA.

Please refer to the attached file my comments.

Author Response

(The authors gave the same response as above.)

Reviewer 3 Report

In the present manuscript, the authors have developed an NEC score system for BLCA patients based on the levels of necroptosis-related genes that can be used for evaluating prognosis, immune infiltration, and immunotherapy responses. However, the major concern is the lack of in-vivo data to test whether the computational findings hold true. The manuscript also needs careful copy editing.  

Author Response

(The authors gave the same response as above.)

Round 2

Reviewer 3 Report

The authors have not performed important experiments to justify the claims.